# Hybrid Precision Floating-Point (HPFP) Selection to Optimize Hardware-Constrained Accelerator for CNN Training

**DOI:** 10.3390/s24072145

**Published:** 2024-03-27

**Authors:** Muhammad Junaid, Hayotjon Aliev, SangBo Park, HyungWon Kim, Hoyoung Yoo, Sanghoon Sim

**Affiliations:** 1Department of Electronics, College of Electrical and Computer Engineering, Chungbuk National University, Cheongju 28644, Republic of Korea; junaid@chungbuk.ac.kr (M.J.); hayotjon@chungbuk.ac.kr (H.A.); sangbopark@chungbuk.ac.kr (S.P.); hwkim@chungbuk.ac.kr (H.K.); 2Department of Electronics, Chungnam National University, Daejeon 34134, Republic of Korea; hyyoo@cnu.ac.kr

**Keywords:** floating points, BFP, MSFP, HPFP, deep neural network (DNN), Yolov2

## Abstract

The rapid advancement in AI requires efficient accelerators for training on edge devices, which often face challenges related to the high hardware costs of floating-point arithmetic operations. To tackle these problems, efficient floating-point formats inspired by block floating-point (BFP), such as Microsoft Floating Point (MSFP) and FlexBlock (FB), are emerging. However, they have limited dynamic range and precision for the smaller magnitude values within a block due to the shared exponent. This limits the BFP’s ability to train deep neural networks (DNNs) with diverse datasets. This paper introduces the hybrid precision (HPFP) selection algorithms, designed to systematically reduce precision and implement hybrid precision strategies, thereby balancing layer-wise arithmetic operations and data path precision to address the shortcomings of traditional floating-point formats. Reducing the data bit width with HPFP allows more read/write operations from memory per cycle, thereby decreasing off-chip data access and the size of on-chip memories. Unlike traditional reduced precision formats that use BFP for calculating partial sums and accumulating those partial sums in 32-bit Floating Point (FP32), HPFP leads to significant hardware savings by performing all multiply and accumulate operations in reduced floating-point format. For evaluation, two training accelerators for the YOLOv2-Tiny model were developed, employing distinct mixed precision strategies, and their performance was benchmarked against an accelerator utilizing a conventional brain floating point of 16 bits (Bfloat16). The HPFP selection, employing 10 bits for the data path of all layers and for the arithmetic of layers requiring low precision, along with 12 bits for layers requiring higher precision, results in a 49.4% reduction in energy consumption and a 37.5% decrease in memory access. This is achieved with only a marginal mean Average Precision (mAP) degradation of 0.8% when compared to an accelerator based on Bfloat16. This comparison demonstrates that the proposed accelerator based on HPFP can be an efficient approach to designing compact and low-power accelerators without sacrificing accuracy.

## 1. Introduction

The interest in deep neural networks (DNNs) from researchers has grown significantly within the last decade due to their tremendous range of applications across numerous spheres. The growing complexity of deep neural networks (DNNs) has significantly increased computational costs and the challenge of implementing these models [1,2,3,4]. CPUs struggle to provide the needed computing capacity, while GPUs offer enhanced speed with their parallel architecture. Neural Processing Units (NPUs) on System-on-Chips (SoCs) potentially offer greater efficiency than GPUs but are hindered by longer development times and higher costs [5,6]. FPGA-based NPUs, despite being a more feasible option, face challenges in deploying large DNN models due to their limited resources. Consequently, researchers are motivated to find ways to reduce DNN’s computational complexity [7,8,9]. In this direction, the research community is putting much effort into creating more cost-effective and efficient arithmetic units for seamless integration with hardware platforms that typically have constrained computational capabilities [10,11,12]. Much research has focused on optimizing hardware for the inference [13,14] of classification and object detection DNNs and developing training accelerators [15,16] for classification tasks. Still, the lapse in object detection DNN training accelerators remains very persistent. Training DNNs is often nontrivial and may involve considerable computational resources and sophisticated hardware design. This complexity is primarily attributed to the use of floating-point operations [17].

Recently, researchers have designed more practical custom floating-point formats that are alternative replacements for the computationally costly IEEE 754 FP32, for instance, NVidia’s TF-32 [18], Google’s bfloat16 [19], and our previous work [20]. In our prior research, we adopted a mixed floating-point precision approach that utilized a 24-bit custom FP format for layers requiring higher precision, while other layers used bfloat16 hardware primitives. Although these custom floating points’ size and energy consumption were significantly lower than those of the IEEE 32-bit format, they still lag behind fixed-point formats in area and power efficiency for similar data widths.

With the increasing need for efficient DNN training, which is crucial for diverse applications such as self-driving cars and sophisticated security monitoring systems, we have developed the Hybrid Precision Floating-Point (HPFP) selection method. This novel approach maximizes computational efficiency and precision within DNN training processes. The key contributions of our work are as follows:**HPFP Selection Method:** We present a method that optimally selects floating-point precision, tailored for both the exponent and mantissa bit widths. This innovative method asymmetrically maps the exponent range, capitalizing on the full potential of available exponent bits, thereby optimizing computational resources. Our strategy utilizes hybrid precision levels for arithmetic operations within each layer while maintaining a uniform, optimal precision for the data path. This hybrid precision approach markedly enhances computational efficiency and optimizes memory usage in DNN accelerators.**Fixed-Point Logic for Arithmetic Operations:** By extracting the sign, mantissa, and exponent components from HPFP, we leverage fixed-point logic for arithmetic operations like addition, multiplication, and division. This reduces computational complexity and makes it a cost-effective solution for performing arithmetic operations in DNNs.**Development of Parameterized CUDA Libraries:** We developed highly flexible CUDA libraries that convert the IEEE 32-bit floating-point format into any configuration of exponent and mantissa bits and allow for custom exponent range adjustments. We have also created libraries for fundamental deep neural network (DNN) layers, such as convolution, max pooling, batch normalization, and their backward functions, which can utilize these custom floating-point formats for verification. These libraries play a crucial role in validating the efficacy of HPFP-based DNN training and ensuring model robustness before hardware implementation. We made these libraries open source to contribute to the broader research community and foster further innovation. The code is available at https://github.com/JunaidCBNU/CUDA-Libraries-RFFP (accessed on 20 February 2024).

In this paper, Section 2 outlines related work, Section 3 provides context and background for our research and highlights the limitations of existing approaches to training deep neural networks (DNNs), which motivated the development of our Hybrid Range Flex floating point (HPFP). Section 4 is dedicated to presenting our proposed architecture, featuring a detailed discussion of our methodology for identifying the optimal HPFP format. Section 5 compares our architecture to related works, demonstrating its effectiveness and efficiency. Finally, Section 6 summarizes the findings and conclusions of the paper, underscoring the contributions and potential future directions of our research.

## 2. Related Research

Training DNNs is complex and computationally intensive compared to inference; as a result, many researchers have proposed training accelerators [21,22,23] aimed at optimizing hardware efficiency. Recent studies have also explored leveraging the inherent sparsity in feature maps (FMAPs) and weights to enhance performance further [24,25]. SIGMA [26] introduces a training accelerator designed to manage sparsity in GEMM operations by implementing a Benes network, ensuring efficient distribution of computational workload. However, the random scattering of zeros in data and feature maps results in overhead due to the complex dataflow hardware required to obtain valid operands.

Researchers have adopted reduced precision in DNN training to enhance arithmetic density and lower computational overhead. Ref. [27] assessed the impact of employing a half-precision floating-point format for training. However, the limited numerical range provided by their 5-bit exponent resulted in issues such as underflow and overflow during the training process. To address the challenges of computational efficiency and precision, [28] introduced mixed precision training. This approach stores weights, activations, and gradients in FP16 to reduce memory usage and accelerate computation. In contrast, an additional copy of the weights is maintained in FP32 to preserve the precision necessary for effective gradient updates. This strategy mitigates the gradient vanishing problem by applying updates to the FP32 weights. However, it also introduces the need for extra memory to store the FP32 weight copies and incurs additional computational overhead due to the necessity of on-chip FP32-FP16 conversion processes.

To address the dynamic range issue without incurring additional overhead, [29] proposed Bfloat16, which offers the same dynamic range as FP32 (IEEE-754 single precision). The application of Bfloat16 in training neural networks was examined by [30], showing that Bfloat16 achieves numerical performance comparable to that of FP32. Further exploration of Bfloat16 and its variant, Bfloat14, for training on resource-constrained FPGAs was conducted by [31]. Training on Bfloat14 was achieved by reducing the mantissa bits while maintaining the exponent bits identical to those in Bfloat16. Although this strategy preserves the high dynamic range, its reduced mantissa often results in diminished accuracy.

As DNNs expand, researchers persist in searching for strategies to further narrow the data width without significantly impacting accuracy. In this context, optimizing the number of exponent bits becomes essential. Block Floating Point (BFP) has been explored in [32,33] within DNNs to reduce the exponent overhead by sharing the exponent across a block of feature data. This method balances floating point (FP) and integer (INT) formats, offering FP-like precision while keeping a hardware footprint similar to that of INT. Recently, Microsoft introduced a BFP-inspired format called Microsoft Floating Point (MSFP) [34] for low-cost DNN inference. This format is available in two variants: MSFP-12 and MSFP-16. However, these studies primarily rely on a fixed BFP format for all layers, limiting potential performance gains. To improve upon existing BFPs, dynamic precision was exploited in [35,36], which demonstrated an increase in throughput of up to 16x by providing a reduction of 4 bits in inference.

This inspired the development of multi-precision block floating-point (BFP), referred to as FlexBlock (FB), in a previous study [21]. This approach computed feature maps and weight gradients in FB12, while local gradients were computed in FB16/FB24. The training time and hardware resource utilization are significantly reduced using this approach. However, the method faces challenges, primarily due to the need for a substantial number of FP32 adders for accumulating partial sums and the reliance on the FB24 format for 38% of the total processing during backward propagation. A recent study [37] introduced an innovative approach to training with BFP by utilizing a very small block size (i.e., 8) alongside stochastic rounding to mitigate quantization error. Despite these efforts, the overhead associated with managing blocks and transmitting a small number of values with a shared exponent hinders the ability to fully capitalize on the potential reduction in the data path.

## 3. Background

### 3.1. Training Deep Neural Networks (DNNs)

Training DNNs is a fundamental process in computer vision and artificial intelligence, enabling these models to learn from vast amounts of data and make intelligent decisions. At its core, DNN training involves adjusting the weight parameters of its convolutional kernels to minimize the difference between the predicted output and actual output.

#### 3.1.1. Stages of Training DNNs

To train a DNN model, the below stages are required:**Forward Propagation:** In this process, every layer in the network sequentially processes the input data, forwarding the output to the next layer, culminating in the generation of the final output**Loss Calculation:** The discrepancy between the network’s output and the actual output (referred to as the ground truth) is quantified using a loss function.**Backpropagation:** This involves computing the gradient of the loss function and propagating it through each layer of the network utilizing the chain rule to facilitate this calculation.**Weight Update:** Optimization algorithms like Standard Gradient Descent (SGD), Adam, or RMSprop are employed to modify the network weights, aiming to reduce the loss to a minimum.

After updating the weights, the model’s predictive capability is tested through inference. It is then evaluated using metrics such as mean Average Precision (mAP) to determine the necessity for further training. Figure 1 illustrates the general training process employing SGD optimization.

#### 3.1.2. Algorithms for Object Detection DNNs

Object detection plays a pivotal role in computer vision, entailing the identification and localization of objects within images or video frames [38]. This field has seen rapid advancements, mainly due to the development and application of deep learning techniques. Unlike mere image classification algorithms that only identify the type of the primary object within an image, object detection algorithms go a step further by not only classifying multiple objects, but also providing the bounding box coordinates for each identified object. This capability allows them to precisely locate and demarcate each object within the image. Figure 2 shows the existing object detection algorithms of two types. Single-shot detectors analyze the input image in a single pass to simultaneously predict object classes and locations. This efficient and fast approach prevents the need for a separate region proposal phase. In contrast, two-shot detectors operate in two stages [39]. The first stage focuses on region proposal, identifying potential areas in the image that may contain objects. The second stage classifies these proposed regions and refines the location predictions. Although two-shot detectors are typically more accurate, especially in complex scenarios or with objects of varying sizes, they tend to be slower than single-shot detectors due to the additional region proposal step. The choice between single-shot and two-shot detectors hinges on the application’s specific requirements, including the need for speed versus accuracy.

#### 3.1.3. YOLOv2-Tiny: A Lightweight Object Detection Model

In assessing the efficacy of our proposed HPFP format, we selected YOLOv2-Tiny as the benchmark object detection model. YOLOv2-Tiny is a streamlined variant of the original YOLOv2, designed for rapid execution without substantially compromising detection accuracy. It embodies the core YOLO principle of simultaneously predicting numerous bounding boxes and their associated class probabilities in a single pass through the network. This simplified variant diminishes complexity by adopting fewer convolutional layers and filters, substantially lowering the computational demands. Figure 3 delineates the streamlined architecture of YOLOv2-Tiny, featuring a concise assembly of nine layers, initiating with a convolution block at its core. It incorporates the batch normalization technique to improve training stability and enhance model generalization. The model employs Leaky ReLU as its activation function, chosen for its ability to facilitate non-linear learning while preventing the vanishing gradient problem at a reduced hardware cost. In addition, the model employs 2 × 2 MAX pooling steps in the upper layers, strategically incorporated to enhance feature extraction by reducing the spatial size of the representation, thus preserving essential information while reducing data volume. The choice of Stochastic Gradient Descent (SGD) for weight updates is made to capitalize on SGD’s strengths, such as its effectiveness in large-scale data processing for better generalization. While we demonstrate the proposed HPFP method using a YOLOv2-Tiny example optimized for speed in this work, HPFP can be applied to any DNN model.

### 3.2. Exponent Overhead Reduction Methodologies

The precision of floating-point representation plays a pivotal role in balancing the accuracy and the computation efficiency in DNN training. The primary motivation behind reducing exponent overhead is to address the excessive overhead posed by traditional floating-point representations, especially in scenarios requiring high throughput and low power consumption. Reducing the exponent size in numerical representations directly impacts the dynamic range of the numbers that can be encoded. The dynamic range refers to the ratio between the smallest and largest values a numerical system can represent. In floating-point representations, the dynamic range is heavily influenced by the size of the exponent; a reduction of just one bit in the exponent width effectively halves the dynamic range of the numbers that can be represented.

The Block Floating Point (BFP) format emerges as a strategic alternative to mitigate exponent overhead. Unlike the straightforward reduction in the number of exponent bits, BFP enables a group of numbers to share a common exponent while maintaining distinct significands. This shared-exponent approach not only preserves numerical precision to a greater extent, but also enhances computational efficiency. The method of converting FP data into BFP data involves the following steps:**Determine the Maximum Exponent:** Determining the maximum exponent needed to represent the largest number in the block, which will serve as the common exponent.**Mantissa Adjustment:** Normalizing the mantissa of all numbers in the block with respect to the common exponent. This process involves scaling the mantissa to align with the common exponent, ensuring each number retains its proportional value.**Rounding:** Rounding or truncating the mantissa to fit within the available mantissa bit width.

Adopting the BFP format can reduce the storage and computational overhead of managing individual exponents for each number in the data.

### 3.3. Problems of Previous Training Accelerators

Our research comprehensively analyzes previous training accelerators based on Block Floating Point (BFP), exploring the complexities they introduce when training across diverse datasets. Below are a few fundamental limitations that make the use of these formats in DNN training challenging:**Difficulty of balancing block size over efficiency:** The efficiency of BFP is highly dependent on the block size; choosing a small block size may not fully exploit the reduction benefits, while a large block size could sacrifice the precision of smaller data values in the common block, limiting overall accuracy.**High computational overhead:** While the BFP approaches utilized highly efficient hardware for BFP dot products and calculating the partial sums, accumulating these partial sums still relies on floating-point arithmetic. Reference [19], for instance, employed 32-bit floating-point arithmetic to accumulate partial sums. Smaller block sizes are needed when dealing with floating-point data featuring a diverse dynamic range, leading to increased floating-point adders. The BFP approaches, therefore, often require on-chip FP-BFP and BFP-FP converters, leading to high computational overhead, which can make it impractical for large DNNs.**Limited dynamic range:** In BFP formats, all data elements in a block share the same (maximum) exponent. This shared exponent approach can restrict the dynamic range of other elements, potentially leading to precision loss for smaller magnitude elements and an increase in quantization errors.

The limitations of these BFP-based formats have inspired us to develop HPFP; this is a step forward in achieving more hardware-optimized architecture, and is explained in detail in the next section.

## 4. Proposed Method

This section outlines an approach to optimizing deep learning models through precision management, structured around three algorithms and integrating a hybrid precision computational unit. Algorithms 1 and 2 focus on identifying the most compact configuration that meets a specified threshold for mean Average Precision (mAP) degradation. Leveraging the groundwork laid by these algorithms, Algorithm 3 advances the optimization process by fine-tuning arithmetic precision on a layer-by-layer basis. This fine-tuning process is complemented by individually optimizing the precision of each data path hardware block for each individual layer, ensuring a balanced approach to enhancing computational efficiency and model accuracy.

**Algorithm 1:** Find Optimal Exponent Bits
**Input:**
•  Activations and Weights of all Layers in IEEE_754 format     {1B Sign, 8B Exponent, 23B Mantissa}•  Epochs, Layers, IEEE_754_Bias = 127, TF_EXP_
**Output:**
•  Activations and Weights of all Layers in Optimal Exponent Format     {1B Sign, [Optimal_Exp_Bits-1:0] Mapped_Exp, 23B Mantissa}1.  Initialize an array ExponentFrequency to track the frequency of each exponent.2.  Set MaxExponent = −∞ and MinExponent = ∞.3.  For epoch = 0 to NumEpochs-1 do:    3.1. For L = 0 to NumLayers-1 do:     3.1.1. For each exponent E in the current layer L of the epoch:       3.1.1.1 Increment ExponentFrequency[E] by 1.     3.1.2. End for.    3.2. End for.4.  End for. 5.  TotalExponents = sum(ExponentFrequency[E]) for all E in all layers6.  For each exponent, frequency in ExponentFrequency do:    6.1. MaxExponent = max(MaxExponent, exponent).     6.2. If frequency/TotalExponents < TF_EXP_, then continue.     6.3. MinExponent = min(MinExponent, exponent).7.  End for.8.  DistinctExponents = MaxExponent − MinExponent.9.  Optimal_Exp_Bits = ⌈log2(DistinctExponents + 1)⌉.10.   TotalCombinations = 2^Required_Exp_Bits.11.   SafetyMargin= TotalCombinations − (DistinctExponents + 1).12.   SafetyMarginEachSide = SafetyMargin/2.13.   Extended Minimum Exponent = MinExponent − SafetyMarginEachSide. # To prevent underflow14.   Extended Maximum Exponent = MaxExponent + SafetyMarginEachSide. # To prevent overflow15.   HPFP_Bias = Extended Minimum Exponent. # HPFP Bias will shift the exp range asymmetrically16.   Mapped_Exp = (Exponent − IEEE_754_Bias) + HPFP_Bias. # Exponent Mapping using HPFP Bias17.   Return the Optimal exponent bits.

**Algorithm 2:** Find Optimal Mantissa Bits 
**Input:**
•  Activations and Weights of all Layers in Optimal Exponent format     {1B Sign, [Optimal_Exp_Bits-1:0]B Mapped_Exp, 23B Mantissa}•  mAP_D_TH_ (Threshold for acceptable mAP degradation)•  AR_TH_ (Threshold for Adaptive Reduction)
**Output:**
•  Activations and Weights of all Layers in HPFP format    {1B Sign, [Optimal_Exp_Bits-1:0] Mapped_Exp, [NewMantissaWidth-1:0] Mantissa}
1.  Initialize OptimalMantissaWidth to 23.2.  Initialize ReductionFactor to some initial value, e.g., 2.3.  Initialize AdjustmentFactor to some initial value, e.g., 0.054.  Initialize mAP_Degradation to 0.5.  While OptimalMantissaWidth > 1 bit do:     5.1. Calculate OptimalMantissaWidth = floor(OptimalMantissaWidth/ReductionFactor).     5.2. Use OptimalMantissaWidth for activations and weights in all layers.     5.3. Apply rounding and overflow checks.     5.4. Calculate mAP_Degradation.     5.5. If mAP_Degradation < mAP_D_TH_,       5.5.1. Adjust ReductionFactor for more aggressive reduction.          ReductionFactor += AdjustmentFactor      5.6. Else       5.6.1. If mAP_Degradation ≥ AR_TH_,         5.6.1.1 Adjust ReductionFactor for a less aggressive reduction. // Moderate           ReductionFactor −= AdjustmentFactor       5.6.2. Else, exit the loop and proceed to fine-tuning.6.  End While.7.  Fine-tuning phase:     7.1. While mAP_Degradation ≥ mAP_D_TH_
      7.1.1. Increment OptimalMantissaWidth by 1.      7.1.2. Use OptimalMantissaWidth for activations and weights in all layers.      7.1.3. Calculate mAP_Degradation.      7.1.4. If mAP_Degradation < mAP_D_TH_, confirm OptimalMantissaWidth and exit the loop.8.  End the Fine-tuning phase.9.  Return the determined OptimalMantissaWidth for HPFP format.

**Algorithm 3:** Algorithm: Layer-wise Precision Optimization 
**Input:**
•  Activations and Weights of all Layers •  Improvement_TH_, Layers•  Precision_levels: Ordered set {p_1_, p_2_, …, p_n_} in ascending order of precision levels
**Output:**
•  Optimal precision configuration for each layer of the DNN model.
1.  Initialize an array of layer_impact_scores to store the mAP scores for each layer2.  Set all layers data path and arithmetic to minimum precision level p_1_
3.  Calculate the baseline mAP (mAP_base_) with all layers set to minimum precision4.  For layer L = 0 to Layers-1 do: // Layers means the total number of layers in the CNN model  4.1. Set the layer *L*’s arithmetic precision to the maximum level (*p_n_*)   4.2. Calculate mAP for the current configuration.  4.3. Assign the mAP value to layer_impact_scores[*L*]  4.4. Reset layer *L*’s arithmetic precision back to *p*_1_
5.  End for6.  Sort layers based on layer_impact_scores in descending order // Prioritize the layers according to a higher mAP impact7.  Set all layers’ arithmetic precision back to *p_1_*8.  For each layer in the sorted list of layers based on their layer_impact_scores[*L*]:   8.1. For *i* = 1 to *n:* // iterate through Precision_levels using index *i*      8.1.1. Increase the arithmetic precision of the current layer to the next level *p_(i+1)_*      8.1.2. Calculate the new mAP (mAP*_i_*) with this new precision setting of *p_(i+1)_*      8.1.3. Calculate improvement = mAP_i_ − mAP_base_
      8.1.4. if improvement ≥ *Improvement_TH_*
       8.1.4.1. mAP_base_ = mAP_i._ (Update baseline mAP to current mAP)      8.1.5. Else:       8.1.5.1. Revert *precision_i_* to *p_i_ from p_(i+1)_*       8.1.5.2. Break the loop // Stop increasing arithmetic precision for this layer9.  End for10.   Return the optimal arithmetic precision configuration for each layer with the datapath precision maintained at the minimum level (p_1_).

### 4.1. Algorithms for Optimal Exponent and Mantissa Bit Widths

We have developed a systematic algorithm to analyze training data, activations, and weights during forward propagation and their gradients during backward propagation to determine the optimal bit width required for floating-point representation. Algorithm 1 shows the procedure of determining the minimum number of bits for the required exponent for a given training DNN model.

**Frequency Analysis (Lines 1~4):** This crucial initial step involves a comprehensive analysis of the exponents’ distribution across all DNN layers for activations, weights, and gradients. By tracking the frequency of occurrence for each exponent, the algorithm identifies the range of exponent values actively used during the model’s training. This analysis helps in recognizing outlier exponents that occur infrequently and may not significantly impact the model’s mean Average Precision (mAP).**Maximum Exponent Identification (Lines 5~6):** The algorithm then determines the maximum exponent.**Minimum Exponent Determination (Lines 5~6):** To ascertain the minimum exponent, the algorithm compares the frequency of each exponent with the Exponent Threshold Frequency (TF_EXP_), which is determined through an evaluation of its impact on the model’s mean Average Precision (mAP). If the minimum exponent across all layers exceeds TF_EXP_, that local minimum exponent is designated as the global exponent for all layers. Otherwise, the algorithm selects the next minimum exponent that exceeds (TF_EXP_).**Optimal Bit Width Calculation (Lines 8~9):** Utilizing the maximum and minimum exponent algorithm effectively delineates the range of unique exponents. Based on this range, the algorithm computes the optimal exponent bit width required to represent these values accurately, ensuring no unnecessary overhead.**Safety Margin Calculation (Line 11):** To ensure robustness, the algorithm calculates the safety margin, which serves as a safety buffer at both ends of the exponent range. This margin is determined by subtracting the count of distinct exponents from the total combinations representable by the optimal bit width.**Extended Exponent Range (Lines 12~14):** The safety margin is equally distributed to extend the minimum and maximum exponent limits, forming the Extended Minimum Exponent (EME) and Extended Maximum Exponent (EME), respectively. This extension is a precautionary measure to prevent underflow (gradient vanishing) and overflow (NaN/Inf) during computations.**Asymmetric Mapping with HPFP Bias (Lines 15~16):** This step optimizes the exponent representation in activations, weights, and gradients according to their requirements. In the final step, the algorithm asymmetrically maps the extended exponent range to the digital scale, using the Extended Minimum Exponent as the HPFP Bias. This mapping optimizes the representation of exponents in activations, weights, and gradients according to their precise needs during training. The HPFP Bias adjustment aligns the exponent range with the DNN model’s actual data requirements, thereby optimizing computational resources and enhancing training efficiency.

This process ensures more efficient use of exponent bits by aligning the exponent range with the actual data needs of DNN training, optimizing computational resources, and improving training efficiency.

While Algorithm 1 determines the minimal exponents’ bit width, Algorithm 2 adopts a strategic approach to optimizing mantissa bit width by leveraging an innovative adaptive reduction factor method. This method expedites the search for a computationally efficient configuration.

**Process Initialization (Lines 1~4):** The algorithm begins by setting the initial mantissa width to 23 bits. It then initializes a reduction factor to a value, for instance, 2, and an adjustment factor to 0.05, and then sets the mAP degradation to zero. These initial values lay the groundwork for the adaptive reduction process.**Mantissa Width Reduction Loop (Lines 5~6):** The algorithm’s core operates in a loop, reducing the mantissa width iteratively based on the reduction factor. It recalculates the mantissa width for each iteration, applies this new width to all layers, and performs rounding and overflow checks. It then calculates the reduction factor resulting from this reduction.
**Aggressive Reduction:** If mAP degradation is below the acceptable threshold (mAP_D_TH_), it indicates room for a more aggressive reduction, and the algorithm increases the reduction factor by the reduction (5.5.1).**Moderate Reduction:** Conversely, if mAP degradation exceeds the Adaptive Reduction Threshold (AR_TH_), suggesting a potentially detrimental impact on accuracy, the algorithm decreases the reduction factor (5.6.1.1). This moderation aims to find a more refined balance by reducing the aggressiveness of further reductions.
**Fine-Tuning Phase (Lines 7~8):** Upon reaching a juncture where further reduction either breaches mAP_D_TH_ or suggests over-reduction, the algorithm halts the reduction loop (5.6.2). If necessary, it then enters a fine-tuning phase where the mantissa width is gradually increased to improve accuracy until the reduction factor falls below mAP_D_TH_ (7, 7.1.1–7.1.4).

Finally, the algorithm identifies an optimal mantissa width that balances computational efficiency and model accuracy.

#### Analysis of HPFP Selection for Example CNN Model

To demonstrate how the Hybrid Precision Floating-Point (HPFP) method balances arithmetic and data path precision, Algorithms 1 and 2 are applied to the YOLOv2-Tiny training model. This example implementation provides critical insights into how the proposed method produces more optimal HPFP formats, laying the groundwork for identifying precision requirements that balance computational efficiency with model accuracy. In the training process in Algorithms 1 and 2, each training iteration for the YOLOv2 Tiny model is conducted using the PASCAL Visual Object Classes (VOC) dataset for 300 epochs. Figure 4a–d illustrates the maximum and minimum exponents of activations, weights, and their gradients across all layers. As shown in Figure 4a, Algorithm 1 analyzes all activations and determines that the maximum exponent is 131, corresponding to +4 when adjusted for the IEEE 754 bias of 127 in Layer 8 during the 18th epoch. Conversely, the minimum exponent of −19 is determined after accounting for the bias. For weights, Algorithm 1 determines the maximum exponent of −2 and the minimum exponent of −2, as depicted in Figure 4b. The analysis in Figure 4b,c demonstrates the exponents of maximum and minimum input and weight gradients. Algorithm 1 disregards extremely small exponent values that occur less frequently than *TF_EXP_*, as they have a negligible impact. For instance, as illustrated in Figure 4d, a weight gradient exponent of −97 appears only once and is thus excluded as an outlier.

Figure 4a–d represent the distribution of exponents in activations, weights, input gradients, and weight gradients, respectively. To accurately represent the 24 exponent values ranging from −19 to +4 shown in Figure 4a, Algorithm 1 designates 5 bits to the activation exponent, representing up to 32 unique exponents. To safeguard against underflow and overflow, Algorithm 1 tactically utilizes the unused 8 values to expand the exponent range as a safety margin to prevent underflow and overflow. It expands the lower and upper limits of the exponent by 4 values, respectively. Similarly, for weights, 5 bits are allocated for exponents within a tailored range from −29 to +2 based on the maximum and minimum exponents of weights.

Weight gradients, encompassing 60 unique exponents from −57 to −3, are allocated 6 bits, representing up to 64 values. Incorporating a safety margin of 2 at both ends, the revised exponent range for weight gradients extends from −59 to −1. The approach for input gradients mirrors that of weight gradients, employing the same adjustment strategy for the exponent range.

These revised ranges for activations, weights, and gradients are then linearly mapped onto a digital scale. Using a 6-bit exponent bias, the traditional mapping approach encompasses a symmetric range from −32 to 31, whereas a 5-bit exponent bias spans from −16 to 15. The conventional mapping technique is limited by its symmetric range around zero, which may not align with the specific demands of varied data. However, utilizing the HPFP Bias, we shift the exponent to an asymmetric range to cover the required exponent values using minimal bits. This adjustment allows for the coverage of necessary exponent values with fewer bits, optimizing the representation efficiency. Figure 5 illustrates how the exponents are mapped on different ranges using the HPFP Bias, demonstrating the effectiveness of this approach in accommodating specific data needs.

After mapping the exponents asymmetrically, Algorithm 2 reduces the mantissa width. For the example presented in this work, we choose the mAP degradation threshold, mAP_D_TH_, of 2%. The algorithm identifies HPFP10 as the minimum precision format that satisfies this threshold. In this precision format, the exponents for activations and weights are set to 5 bits; for gradients, it is 6 bits. The mantissa width for activations, weights, and their gradients is set to 3 bits. Given that HPFP10 is the minimum, any HPFP format larger than HPFP10, with a wider mantissa, would also fulfill the degradation threshold.

### 4.2. Layer-Wise Precision Optimization

In the pursuit of an optimal balance between a CNN model’s accuracy and computational efficiency, we present Algorithm 3 for layer-wise precision optimization, an algorithm conducted after Algorithms 1 and 2. Traditional approaches to layer-wise precision optimization face a notable challenge: accelerator hardware is usually implemented to support the maximum precision among all the optimal precisions of individual layers. Such hardware implementation often negates the potential advantages of layer-wise precisions optimization that can minimize the power consumption and memory access overhead, leading to increased memory demands to accommodate the highest precision levels.

To overcome this limitation, we adopted memory access time (MAT) as our primary metric for evaluating the DRAM memory access efficiency of hybrid precision floating-point (FP) formats. MAT quantifies the time required to access memory for each FP format while processing a batch of eight images. Figure 6 illustrates the memory access times, normalized against the minimum configuration format identified by Algorithms 1 and 2. Since the DRAM access time is directly proportional to the number of bits in the data path, our algorithm uses the minimum precision level for the data path to maximize memory access efficiency. We effectively balance the computational accuracy against the hardware and memory usage optimization by enabling layer-wise precision adjustment for arithmetic operations while further reducing the precision of the individual data path operator hardware.

Drawing on the fundamental principles in Algorithms 1 and 2, Algorithm 3 employs a sorted hierarchy of precision characterized by specific widths for both exponents in forward and backward propagation and for the mantissa. This strategy is deployed to enhance computational efficiency by dynamically adjusting the mantissa width for individual layers within a deep neural network (DNN), thereby refining the model’s precision management. Algorithm 3 is predicated on the insight that not all layers require an equal width of exponent and mantissa to satisfy target accuracy. By evaluating the impact of each layer’s precision on the model’s mAP, Algorithm 3 selectively applies higher precision to layers with a more significant effect on accuracy while allocating lower precision levels to less impactful layers. Algorithm 3 mainly consists of eight steps, as explained below:**Initialize an array “layer impact scores” (Line 1):** This array stores the mAP scores for each layer when set to the maximum precision level. This step prepares for assessing how each layer’s precision affects the overall model’s mAP.**Set all layers to minimum precision level (Line 2):** Initially, all layer’s data paths and arithmetic are set to the lowest precision level (p_1_) to establish a baseline for comparison. Lower precision levels generally lead to faster computations but might compromise the accuracy of results, making it essential to find the optimal balance.**Calculate the baseline mAP (Line 3):** Determine the model’s performance at the initial, lowest precision configuration to serve as a reference for improvement measurement.**Iterate over each layer (lines 4 to 5):** For every layer in the DNN, perform the following:
Set the layer’s arithmetic precision to the maximum level (p_n_) to evaluate its potential impact on the model’s performance.Calculate the mAP for the current configuration, with the specific layer arithmetic at p_n_ and the rest at p_1_.Store the calculated mAP in layer impact scores, associating the score (mAP) with the layer’s index.Reset the layer’s arithmetic precision back to p_1_ after evaluation.
**Sort layer impact scores and layer indices (Line 6):** Arrange the layers in descending order of their impact scores. The sorting ensures that layers with the greatest potential impact (higher mAP) on performance are prioritized for precision optimization.**Reinitialize all layers to minimum precision level (Line 7):** Before beginning the incremental optimization process, all layers are reset to the minimum precision level. This step ensures the model is in a baseline state, allowing for a systematic increase in arithmetic precision. By incrementally enhancing precision based on the prioritized impact of each layer, the approach efficiently targets precision increases to areas that yield significant performance improvements.**Iterate over each layer in sorted order of impact (lines 8 to 9):** For the layers sorted by their potential impact, gradually increase their arithmetic precision and evaluate the performance gain:
Increase the precision of each arithmetic block in the current layer to the next level p_(i+1)_.Calculate the mAP with the updated precision for the layer.Determine the improvement in mAP compared to the baseline mAP.If the improvement meets or exceeds a predefined improvement threshold, update the baseline mAP to this new mAP value and test the next higher arithmetic precision level for the layer.If the improvement does not meet the threshold, revert the layer’s arithmetic precision to the previous level and stop increasing precision for this layer.
**Return the optimal precision configuration (Line 10):** After iterating through all layers and adjusting their arithmetic precision levels based on the improvement threshold, the final configuration is deemed optimal for the DNN model. The configuration has a data path set at p_1_, while arithmetic is optimized against each layer.

To demonstrate Algorithm 3, two experiments were conducted using the example CNN of YOLOv2-Tiny to optimize arithmetic precision, setting an improvement threshold of 0.1. Suppose that Algorithms 1 and 2 determine HPFP10 as the minimum precision, which is applied to the precision of all the data paths in all layers.

**Two-Precision-Level Optimization:** In the case where Algorithms 1 and 2 choose two precision levels (HPFP10, HPFP12) as input, many experiments with Algorithm 3 show a trend where lower layers closer to the output often require higher arithmetic precision than upper layers closer to the input. Specifically, as illustrated in Figure 7, the algorithm assigns the lowest precision level, HPFP10, to the arithmetic of initial layers (0–3), while the higher precision level, HPFP12, is allocated to the upper layers (4–8) during forward propagation. HPFP12 is determined to be the optimal arithmetic precision for all layers for backward propagation.**Three-Precision-Level Optimization:** In the case where Algorithms 1 and 2 choose three precision levels (HPFP10, HPFP12, HPFP14) as input, Algorithm 3 assigns HPFP10, HPFP12, and HPFP14 to different layer’s arithmetic, as shown in Figure 8.

Figure 9 depicts the mean Average Precision (mAP) degradation observed across various floating-point (FP) format implementations for the YOLOv2-Tiny model. Each FP format is categorized based on the exponent size in forward data, backward data, and the size of the mantissa. Notably, Bfloat16 and HPFP-14 exhibit the same level of mAP degradation, which is attributed to their identical mantissa widths. In contrast, HPFP10 and HPFP12 undergo mAP degradations of 0.72% and 1.8%, respectively. Employing a two-precision-level optimization strategy with example precision levels (HPFP10, HPFP12) leads to a mAP degradation of 1.2% compared to FP32 while using a three-precision-level optimization strategy with example precisions (HPFP10, HPFP12, HPFP14) results in a mAP degradation of 0.96% compared to FP32. A more detailed comparison is provided in Section 5.2, Table 1.

### 4.3. HPFP Multiplication and Accumulation (HPFP MAC)

A Hybrid Precision FP MAC (HP-MAC) unit is presented as an example implementation of an accelerator for YOLOv2-Tiny, which consists of 3 × 3 convolution kernels in all nine layers based on the diagonal cyclic array proposed by [40]. The input activations are propagated horizontally through each row, while weight parameters are propagated vertically through each column of the 3 × 3 array, as illustrated in Figure 10.

To implement the HP-MAC, traditional adders are utilized with minimal modifications, where the mantissa is adjusted according to the exponent difference before the accumulator operator. However, accommodating the reconfigurable exponent bit width supporting HPFP necessitates developing a specialized multiplier, as depicted in Figure 11. An additional input distinguishes this multiplier, HPFP Bias, which indicates a shifted range for the exponent. This bias is configured through the host and sent to instruction memory as a configuration parameter. The form separator separates the sign, exponent, and mantissa to process them individually. After adding the exponents of both input numbers, the bias adjustor subtracts the HPFP Bias from the sum of the exponents. As determined in Section 4.2, it is noteworthy that the exponent range of activations differs from that of weights in the considered example. In this case, the weight bias will be considered HPFP Bias. By subtracting this bias, we can align the resultant activation within the range of activations, thus eliminating the need for additional overhead. The sign determination employs a two-input XOR gate. Regarding the mantissa, we append the implicit leading 1 to the MSB of each mantissa operand to maintain the normalized floating-point format, proceeding with their integer multiplication. The exception handler handles the edge cases, e.g., NaN (Not a number), infinity, and zero. To ensure the output mantissa matches the input mantissa width, a normalizer right-shifts the mantissa result, rounds it, and correspondingly increments the exponent.

### 4.4. Overall DNN Architecture

To comprehensively evaluate the performance of our proposed Hybrid Precision Floating-Point format, we designed a deep neural network (DNN) accelerator using the proposed hybrid HPFP arithmetic that facilitates inference and training for the YOLOv2-Tiny model. Figure 12 illustrates the architecture of an example DNN training accelerator, as reported in [41].

The training operation of the accelerator begins with the host CPU sending Microcode (instruction set), Input FMAP (images), and weights to DDR DRAM through the PCIe Interface managed by the Direct Memory Access (DMA) controller. Following this, the host CPU configures the DRAM base address for the Microcode and its length and a Read Enable signal for the DNN Controller through the AXI bus. Once the Microcode is read from DDR DRAM via the AXI bus, each layer is processed based on the Microcode instructions. In the accelerator design based on the hybrid HPFP, a lower precision format, HPFP-10, is employed in most computations, including multiplication. A higher precision format, HPFP-12, is employed only for accumulation to ensure more accurate results. It captures finer details in the computational process that HPFP-10 might miss due to its limited precision. Subsequently, the results are normalized back to HPFP-10 to align with our system’s data path based on HPFP-10. These normalized results are then advanced to the Post-Convolution Processing Unit (PCU). In the forward pass, the PCU performs critical functions such as batch normalization, activation, and pooling after the convolutional block processes the multiply and accumulate operations. Conversely, during the backward propagation phase, the operational sequence is reversed: the PCU’s functions are executed in reverse order before convolution, handling the calculation of activation and weight gradients. Following the backward propagation, weights are meticulously updated for each batch, optimizing the system to achieve the optimal mean Average Precision (mAP). After processing, the output feature maps are stored in DDR DRAM through the AXI bus. This strategic use of hybrid precision formats within the HP-MAC and subsequent data normalization underscores the key advantage of the proposed approach, which can balance computational precision and data path efficiency.

The loss calculation is executed on the host CPU instead of adding a custom hardware block, which maximizes the benefit of hardware/software co-design. This strategic hardware/software co-design allows for dynamic adjustments and optimization during the training process, such as employing various optimizers for weight updates or adjusting learning rates based on performance metrics, optimizing the training cycle for improved model accuracy. In addition, the presented accelerator design can accommodate various CNN models thanks to the high flexibility offered by the Microcode-based controller and the hardware/software co-design. However, this work shows the result of the YOLOv2-Tiny model for simplicity.

## 5. Evaluation

### 5.1. FPGA Implementation

To thoroughly evaluate the benefits and measure the actual performance of HPFP formats, we implemented the example CNN training accelerator on a state-of-the-art Virtex Ultra Scale+ VCU118 FPGA platform. This FPGA platform features 1182.2 K lookup tables (LUTs), 2364.5 K flip-flops (FFs), 2160 36 KB of Block RAMs (BRAMs), and 6840 Digital Signal Processing (DSP) slices, alongside six Peripheral Component Interconnect Express (PCIe) lanes. For the hardware/software co-design operations, the FPGA platform interfaces with a host CPU through a PCIe 2.0 connection utilizing one lane. A PC is used for the host CPU and operates with an Ubuntu 20.04 LTS 64-bit environment. Figure 13 illustrates the hardware configuration of the FPGA platform.

The FPGA implementation of the example accelerator using HPFP consumes 255 K LUTs, representing 21% of the total LUT capacity of the FPGA. This utilization showcases our ability to leverage the FPGA’s resources to efficiently achieve high performance. Additionally, it utilizes 314k FFs, 198 BRAMs (a utilization of 13.2%), and 2236 DSP slices (a utilization of 32.7%). The accelerator operates at a frequency of 250 MHz. The low utilization of the FPGA’s resources demonstrates the advantage of the HPFP-based accelerator in achieving optimal performance through efficient hardware utilization, paving the way for future enhancements and applications of our accelerator.

### 5.2. Comparison with Other Works

Due to limited work on training object detection models, it is difficult to directly compare the hardware cost and power efficiency with the previously published works. We present two comparative experiments: one comparing our accelerator for an object detection CNN (YOLOv2-Tiny) and a classification algorithm CNN using the MNIST dataset with previous accelerators for classification CNNs (Resnet20 and Vgg16), and the other comparing three implementations of our accelerators for YOLOv2-Tiny based on the following configurations:Brain floating-point Bfloat16 for both data path and arithmeticA three-precision-level layer-wise configuration, utilizing a data path in HPFP10 with layer-wise arithmetic precision (HPFP10, HPFP12, HPFP14).A two-precision-level layer-wise configuration, employing a data path in HPFP10 with layer-wise arithmetic precision (HPFP10, HPFP12).

Our findings, summarized in Table 1, reveal that the two-precision-level HPFP configuration achieves the lowest power consumption among all the approaches evaluated. Remarkably, despite a modest mAP degradation of 1.2%, this configuration offers a 49.4% reduced energy consumption to process one image in both forward and backward propagation of all layers and a 1.59 times reduction in lookup table (LUT) utilization compared to Bfloat16. These efficiency improvements are attributed to the reduced mantissa width, the implementation of an optimized hybrid precision–MAC (HP-MAC) unit, and an optimal data path. Furthermore, it is 22.35% more energy efficient and consumes 1.36 times fewer LUTs compared to the three-precision-level HPFP configuration, making it highly suitable for scenarios where computational resources are limited or where minimizing energy consumption is crucial. In contrast, the three-precision-level configuration is tailored for applications requiring higher accuracy, with a minimal mAP degradation of just 0.96%.

This evident trade-off between resource efficiency and model accuracy emphasizes the need for innovative solutions to effectively balance these aspects. Our methodology showcases superior resource utilization and energy conservation and opens up new avenues for further research in optimizing hardware design for deep learning applications.

**Table 1 sensors-24-02145-t001:** Comparison with related work.

Criteria	[42]	[20]	[43]	[41]	Proposed		
Year	2021	2022	2023	2024	This Work		
Precision	Fixed 8 (8 bits)	Mixed (16/24 bits)	PINT (8 bits)	Bfloat16 (16 bits)	HPFP ^1^ 3 Hybrid Precision	HPFP ^2,3^ 2 Hybrid Precision	
CNN Model	VGG-16	CNN	ResNet-20	YOLOv2-Tiny	YOLOv2-Tiny	YOLOv2-Tiny	CNN ^4^
Training dataset	CIFAR-10	MNIST	CIFAR-10	Pascal VOC	Pascal VOC	Pascal VOC	MNIST
FPGA Device	VC709	ZCU104	VC709	VU118	VU118	VU118	ZCU104
Clock Frequency		50 MHz	200 MHz	250 MHz	250 MHz	250 MHz	50 MHz
mAP degradation ^5^	-	-	-	0.4%	0.96%	1.20%	-
Num. of LUTs	171,248	33,404	132,000	342,675	292,764	215,916	17,870
Num. of FFs	143,565	61,532	-	79,714	48,930	31,478	32,764
Num. of DSPs	2324	0	1728	2842	2654	2236	0
DRAM Access per image (ms) ^6^	-	-	-	207.70	129.81	129.81	-
Time per image (ms) ^7^	-	0.0134		331.96	254.07	254.07	0.0098
Average Power (W)	16.27	0.635	8.55	8.05	6.85	5.32	0.372
Energy per image (J) ^8^	-	0.0085	-	2.672	1.740	1.351	0.0036

^1^ Format (1B Sign, [5B Exp FW, 6B Exp BW], [3B Man Data Path, Layer Mantissa LM: 3,5,7]). ^2^ YOLOv2-Tiny Format (1B Sign, [5B Exp FW, 6B Exp BW], [3B Man Data Path, Layer Mantissa LM: 3,5]). ^3^ CNN Format (1B Sign, [4B Exp], [5B Man Data Path, Layer Mantissa LM: 5,7]). ^4^ CNN model is based on [18], with 1 Convolution, 1 Pooling and 2 Fully Connected layers. ^5^ mAP degradation: Subtracting mean Average Precision (mAP) of baseline FP32 with other format’s mAP. ^6^ DRAM access time per image during forward and backward propagation across all layers. ^7^ Time per image includes the process time and DRAM access time across all layers. ^8^ Energy consumed per image during forward and backward propagation across all layers.

## 6. Conclusions

This study proposed highly parameterized algorithms for developing a custom floating-point format, namely HPFP, which maps any exponent range based on specific training data and a model into minimum possible exponent bits, and then linearly reduces the mantissa bit to achieve the optimal arithmetic. Our algorithm’s modular design and arithmetic units facilitate multi-precision training, accommodating varying precision requirements across data paths and layers. We implemented our hybrid precision training accelerator on Xilinx VCU118 to evaluate its performance. Experimental results showed power consumption that was 1.5 to 3 times lower than that of previous works. Specifically, in the two-precision-level layer-wise optimization HPFP configuration, adopting the HPFP-10 format for the data path significantly reduces off-chip memory data access by 1.6 times compared to traditional 16-bit FP accelerators and 3.2 times compared to 32-bit FP systems. Employing layer-wise arithmetic operations with HPFP10 and HPFP12 resulted in a minimal accuracy degradation of only 1.2%, demonstrating the efficiency of our approach. In the future, we aim to scale our design across multiple field-programmable gate arrays (FPGAs) to support a more significant number of processing elements (PEs), further enhancing our system’s energy efficiency. This future work will reinforce HPFP’s position as a scalable, precision-flexible solution, significantly enhancing energy and resource efficiency in DNN accelerator designs.

## Figures and Tables

**Figure 1 sensors-24-02145-f001:**
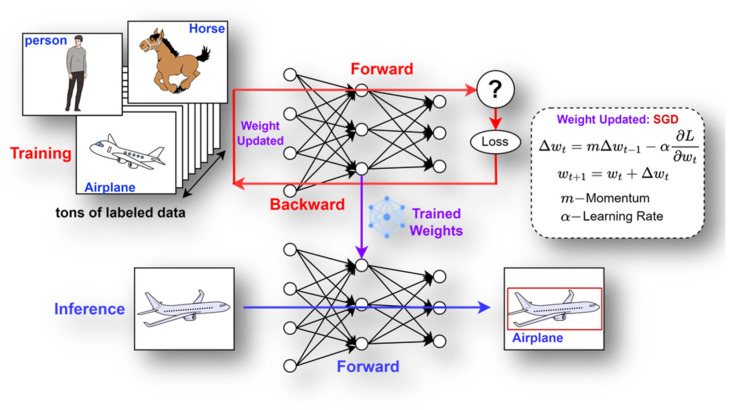
General training process in DNNs.

**Figure 2 sensors-24-02145-f002:**
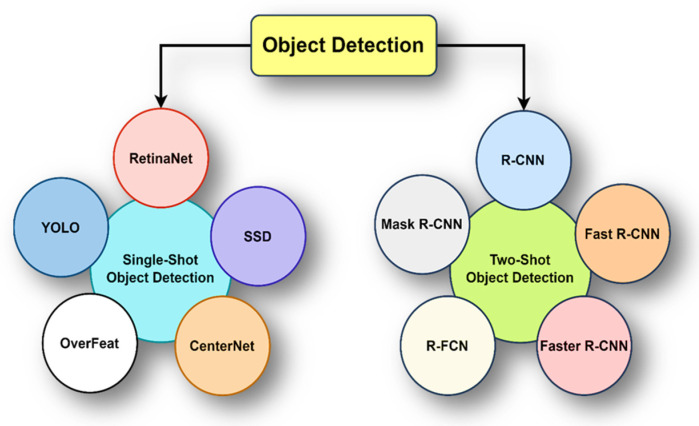
Type of object detection algorithms.

**Figure 3 sensors-24-02145-f003:**
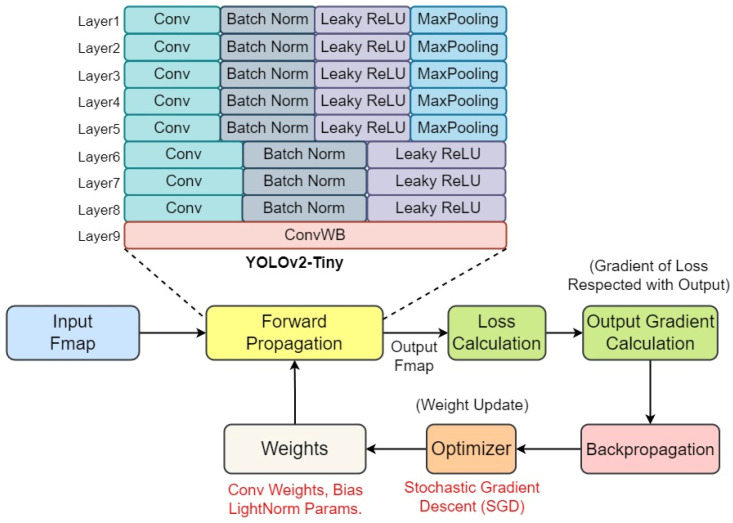
The architecture of the light-weight YOLOv2-Tiny.

**Figure 4 sensors-24-02145-f004:**
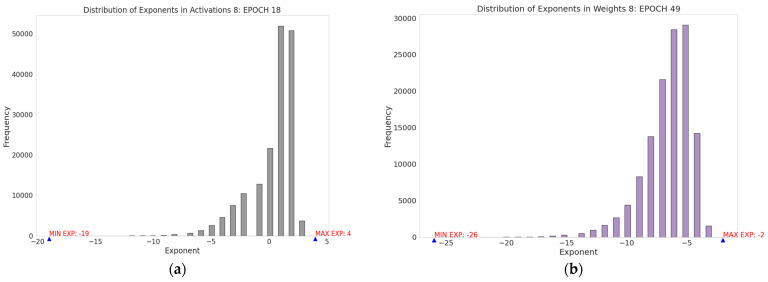
(**a**) Exponent range in activations; (**b**) exponent range in weights; (**c**) exponent range in input gradient; (**d**) exponent range in weight gradient.

**Figure 5 sensors-24-02145-f005:**
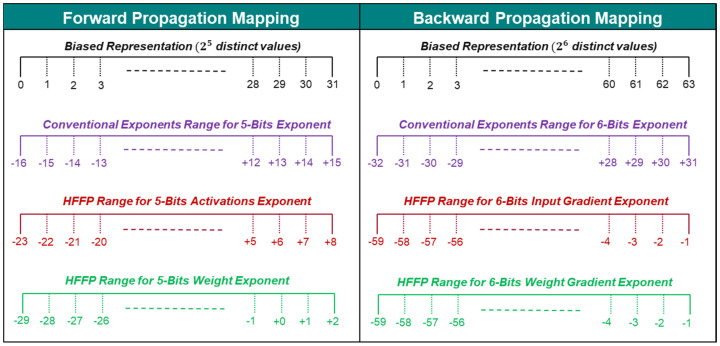
Exponent mapping for activation, weights, and gradients.

**Figure 6 sensors-24-02145-f006:**
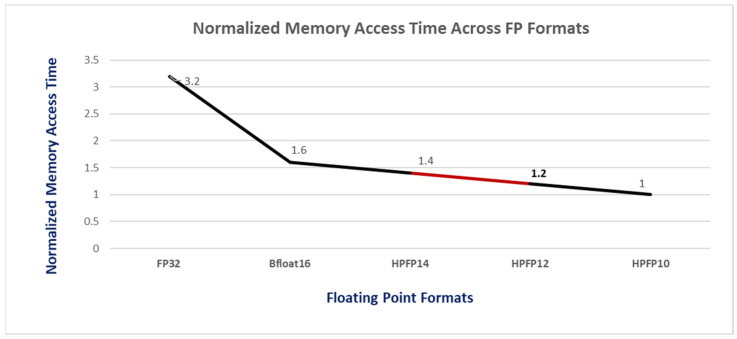
Normalized memory access time across FP formats.

**Figure 7 sensors-24-02145-f007:**
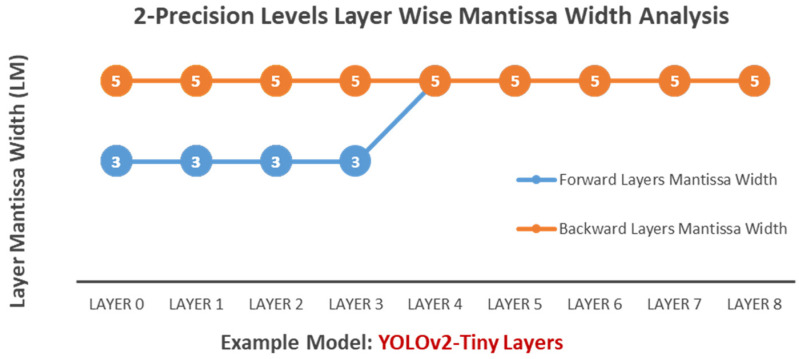
HPFP two-precision-layer-wise configuration example.

**Figure 8 sensors-24-02145-f008:**
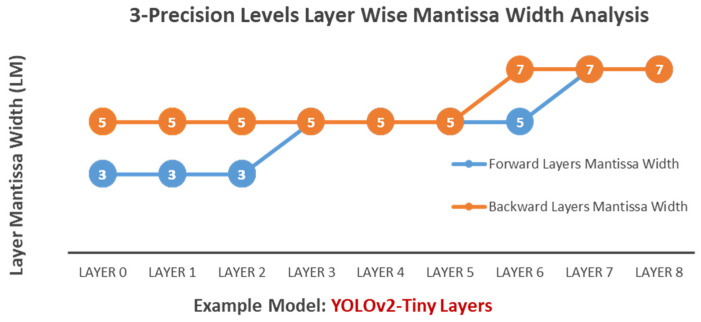
HPFP three-precision-layer-wise configuration example.

**Figure 9 sensors-24-02145-f009:**
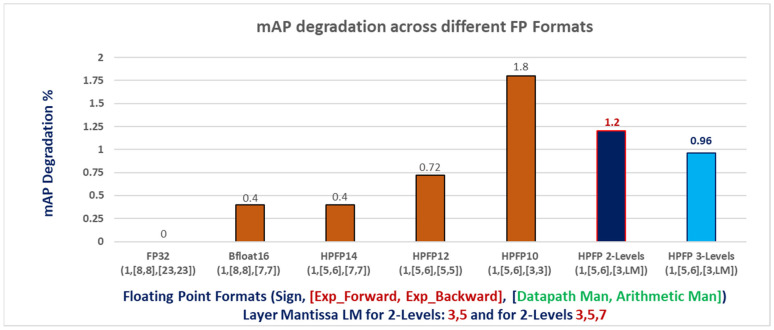
mAP degradation across FP formats.

**Figure 10 sensors-24-02145-f010:**
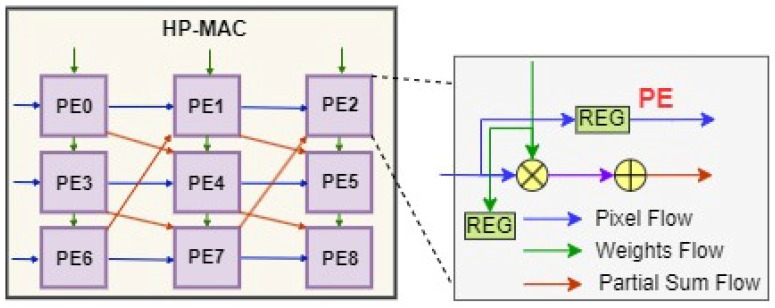
HPFP multiplication and accumulation unit.

**Figure 11 sensors-24-02145-f011:**
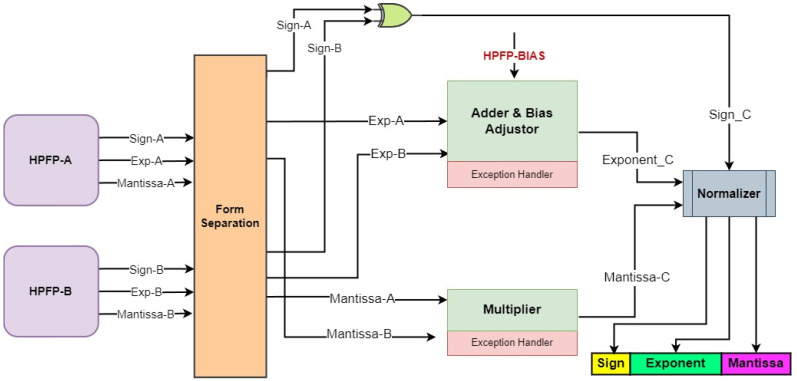
HPFP multiplier.

**Figure 12 sensors-24-02145-f012:**
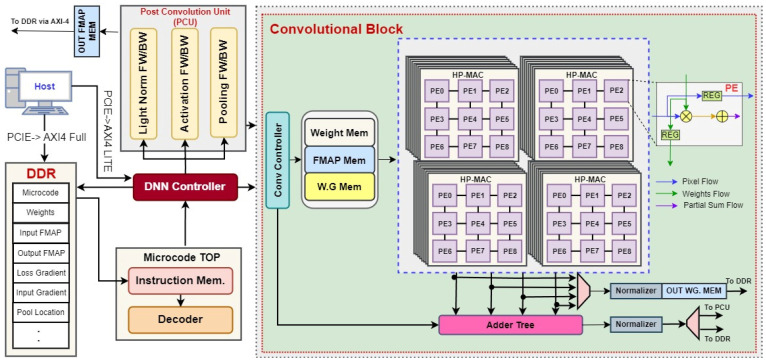
Overall architecture of DNN training accelerator.

**Figure 13 sensors-24-02145-f013:**
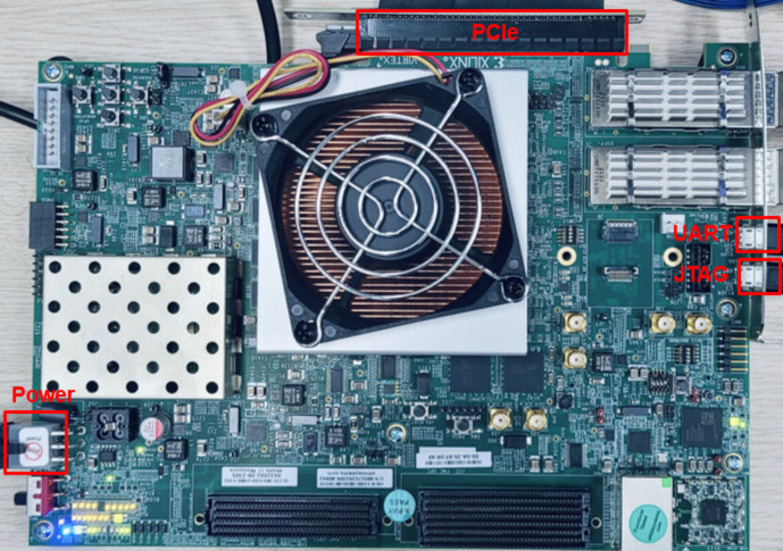
Hardware setup using VCU118.

## Data Availability

Data are contained within the article. (https://github.com/JunaidCBNU/CUDA-Libraries-RFFP accessed on 20 February 2024).

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
