# Peer review of "Hybrid Precision Floating-Point (HPFP) Selection to Optimize Hardware-Constrained Accelerator for CNN Training"

_sensors, 2024, doi:10.3390/s24072145_

Round 1

Reviewer 1 Report

Comments and Suggestions for Authors

The paper presents the results of the research which are dedicated to optimization o the number of bits used to represent floating-point description of the parameters of AI-based detection algorithm. Particularly, reduction of the computer resources required is of concern to accelerate calculations simultaneously keeping their quality at the satisfactory level. The work refers to the known approach of the use of various floating point data formats, accordingly to the desired needs and the specificity of the targeted object being detected/recognized with AI-based algorithms and the strategy of mixed-precision training. The Authors also advantageously plan to make use of CUDA libraries. The newly-presented approach seems to be of great importance, especially for the datasets that exhibit high level of diversity. In the reviewer ‘s opinion the paper is well written and only requires minor editorial corrections before publications.

I am confused, I think that something is mixed regarding the scopes of the subsequent sections. In the end of Section 1, it is written:

-Section 2 “outlines related work” – this is correct, but the background (the context and background for our research) is presented in Section 3

-Section 3 “offers an overview of popular object detection algorithms” – this is correct, but the limitations of existing approaches are also presented in Section 3.3, not in Section 4, as stated by the Authors

- Section 5 “is dedicated to presenting our proposed architecture” – not correct, the content of Section 4 provides this description.

- Section 6 “compares our architecture to related works” – it is not correct, this the content of Section 5.

- Section 7 “summarizes the findings and conclusions of the paper” – it is not correct, this the content of Section 6.

Please make sure the description of the paper content presented in Section 1 is correct.

Flaws and minor issues:

-the title of the section “2. Related” -> “2. Related research”

-row 275 – formatting error, regarding the number of the point “1. The limitations of these”. Please correct it.

-row 311: the letter “T“ in “To” should not be in bold face

-row 403: “algorithm” -> “Algorithm”

-rows 491, 534: “we” -> “the authors” (it is recommended to use passive form)

-row 527: “We present” -> “There was elaborated and presented” or similar form

-row 534: “we utilize” -> “there are utilized”

-row 544: “our example” -> “the considered example” and in the other similar cases, it is recommended to use passive forms

-references: please make sure that all the cited works are of the correct format

Author Response

Thank you for your review, please see the attachment for our response

Reviewer 2 Report

Comments and Suggestions for Authors

1. Objectives are clear

2. Contributions are clear

3. Well described methodology 

3. Sufficient experiment

4. Enough reference

Author Response

(The authors gave the same response as above.)

Reviewer 3 Report

Comments and Suggestions for Authors

Main Contribution:

  • The main contribution of the paper is the introduction of Hybrid Precision (HPFP) selection algorithms, which aim to reduce precision and implement hybrid precision strategies in order to optimize hardware-constrained accelerators for CNN training. This approach allows for a balance between layerwise arithmetic operations and data path precision, resulting in significant hardware savings and improved energy consumption and memory access.

Suggestions for Improving the Paper:

  • Provide more details on the specific algorithms and methodologies used for the HPFP selection process.
  • Include a more comprehensive evaluation of the proposed HPFP approach by testing it on a wider range of CNN models and datasets.
  • Discuss the potential limitations and trade-offs of using reduced precision formats like HPFP, particularly in terms of accuracy and convergence speed.
  • Compare the performance of the proposed HPFP approach with other existing techniques for optimizing hardware-constrained accelerators for CNN training.
  • Give a more thorough analysis of the energy savings and memory access reductions made possible by the HPFP approach, making comparisons with other precision formats.
  • Discuss the potential impact of the HPFP approach on the overall training time and convergence of CNN models.
  • Include a discussion on the potential challenges and considerations for implementing the HPFP approach in real-world edge devices.
  • Give some ideas about where the HPFP approach could go in the future and how it could be used to improve hardware-limited accelerators for AI tasks other than CNN training.
  • The authors are invited to include some recent references, especially those related to  Deep Convolutional Neural Networks.  For instance, the authors may include the following interesting references (and others):

    a. https://www.mdpi.com/2073-431X/12/8/151

    b. https://www.taylorfrancis.com/chapters/edit/10.1201/9781003393030-10/learning-modeling-technique-convolution-neural-networks-online-education-fahad-alahmari-arshi-naim-hamed-alqa
Comments on the Quality of English Language

Can be improved.

Author Response

(The authors gave the same response as above.)

Round 2

Reviewer 3 Report

Comments and Suggestions for Authors

The authors considered my comments and suggestions 

Comments on the Quality of English Language

A final proofread would be useful